# Thermal Modeling of Polyamide 12 Powder in the Selective Laser Sintering Process Using the Discrete Element Method

**DOI:** 10.3390/ma16020753

**Published:** 2023-01-12

**Authors:** Reda Lakraimi, Hamid Abouchadi, Mourad Taha Janan, Abdellah Chehri, Rachid Saadane

**Affiliations:** 1Laboratory of Applied Mechanics and Technologies, ENSAM, Mohammed V University, Rabat 10100, Morocco; 2Department of Mathematics and Computer Science, Royal Military College of Canada, Kingston, ON K7K 7B4, Canada; 3SIRC-LaGeS, Hassania School of Public Works, Casablanca 20000, Morocco

**Keywords:** selective laser sintering, discrete element method, polyamide 12, thermal modeling, additive manufacturing processes

## Abstract

Selective laser sintering (SLS) is one of the key additive manufacturing technologies that can build any complex three-dimensional structure without the use of any special tools. Thermal modeling of this process is required to anticipate the quality of the manufactured parts by assessing the microstructure, residual stresses, and structural deformations of the finished product. This paper proposes a framework for the thermal simulation of the SLS process based on the discrete element method (DEM) and numerically generated in Python. This framework simulates a polyamide 12 (PA12) particle domain to describe the temperature evolution in this domain using simple interaction laws between the DEM particles and considering the exchange of these particles with the boundary planes. The results obtained and the comparison with the literature show that the DEM frame accurately captures the temperature distribution in the domain scanned by the laser. The effect of laser power and projection time on the temperature of PA12 particles is investigated and validated with experimental settings to show the reliability of DEM in simulating powder-based additive manufacturing processes.

## 1. Introduction

Selective laser sintering (SLS), one of the additive manufacturing techniques, uses computer-aided design to directly manufacture solid 3D products by sintering powder layer by layer [1]. This process is one of the additive manufacturing techniques, which is capable of producing exceedingly sophisticated items from a wide variety of simple materials without requiring the use of any tools.

Polyamide, generally known as nylon, is one of the powders that can be utilized in laser sintering. Polyamide is a durable and flexible material resistant to abrasion and wear. It is utilized in numerous industries, including the automotive, aerospace, and medical sectors. Several academic works aiming to study the SLS process have used PA12 powder as their research subject.

Figure 1 is a simplification of each process involved in the manufacturing of SLS. Ref. [1] provides a more in-depth overview of the SLS procedure.

The results of the SLS method include a variety of defects, such as size faults, layer delamination, porosity, and subpar material qualities [2]. These defects could be visible in the finished products. As a result, in order to achieve the goal of regulating the process-related factors and improving the quality of the final product, it is necessary to have a comprehensive understanding of the behavior of the powder while being projected by the laser source.

The behavior of the powder particles is affected by the complicated and variable geometric shapes of the particles, which further adds to the difficulty of the issue. In order to satisfy industrial standards and produce parts of the highest quality, a particle-based numerical modeling framework is required to describe the physical interactions of the powder particles that occur during the SLS process. This is necessary to produce parts that meet the industry’s requirements.

Several different approaches have been created to numerically simulate the physical interactions in the powder bed. These approaches aim to better understand the SLS process, better manage the equipment, and optimize the associated concerns.

The bulk of numerical studies on processes based on powders that are proposed in the literature use continuous methods, which treat the powder bed as a homogenous, continuous medium. This is because continuous methods assume that the powder bed is a single medium. In addition, they use a numerical method for solving multiphysics partial differential equations, which tends to minimize the significance of the role that air plays in the SLS process by acting as a buffer between the particles.

In the work of Gusarov et al. [3], a finite difference approach was used to model the melt. The two-flow method was employed to solve the radiative transfer equation, taking into account the diffusion and conduction heat transfer equations.

Dong et al. [4] used a 3D model with the finite element method (FEM) to study the temperature distribution in the sintering bath during the SLS process. The study considered the latent heat of phase transformation, the effect of convection in the powder bed, and the use of a laser with a power of 8 W at a scanning speed of 0.33 m/s.

Dong et al. [5] used the commercial software Abaqus, which uses the same numerical method to predict the temperature and density distribution of sintered polymers in the powder bed. Fisher et al. [6] also proposed a 2D FEM model of the periodic pulses of a laser on a polymer powder.

However, all cited research works based on continuous methods (FEM, FD, etc.) have limitations when modeling granular media. These models are incapable of capturing the physical behavior of the particles in the powder bed, and they significantly underestimate the effect of the air between the particles and ignore the discontinuity of the simulated material.

Moreover, the physical interactions in the powder bed are caused by collisions between the particles. The interactions are highly dependent on the type of bonding between the particles. Due to the SLS process’ discrete character and discontinuous behavior, modeling with continuous approaches is incompatible with the procedure.

In comparison, the discrete element method (DEM) is a numerical approximation technique used to investigate granular materials. It is capable of simulating a set of particles, investigating their physical reactions based on particle contact laws, and capturing the physical interaction between particles to produce the macro behavior of the simulated medium. The discontinuous medium is treated more realistically by DEM than the continuous method, making it the optimal technique for modeling the SLS process.

Despite all the above advantages, powder bed modeling by DEM is relatively rare in the literature. An exception is a study by Steuben et al. [7], which demonstrated the possibility of using the discrete element method in powder-based additive manufacturing processes with a relatively simple set of approximations and physical assumptions.

This paper develops a framework for thermal modeling powder-based additive manufacturing processes using the discrete element method. This framework describes in detail the steps and calculations required for the thermal simulation of particles in the powder bed.

To validate this DEM framework, it is applied to polyamide 12 (PA12) powder to study the temperature evolution at the center of the laser beam. The results are compared with experimental work by Lanzl et al. [8] and with the results of continuous methods implemented by Yaagoubi et al. [9]. 

In a second study, the effect of laser power and projection time on the temperature evolution of PA12 particles is investigated and compared with the experimental work. This model is implemented numerically in the Python programming language, following the basic steps of DEM.

The discrete element method is described in the following section. Section 3 provides a synopsis of the suggested methodology. Section 3 provides the mathematical model, its implementation, and its architecture. The findings and analysis are presented in Section 4. This paper is concluded in Section 5.

## 2. Discrete Element Method

The granular medium is represented in the discrete element method by a set of particles of simple geometric shape. Generally, spheres are treated independently as solids. These particles interact with each other through the laws of contact and friction between the particles. This discrete method is often used to simulate granular media because it describes their kinematics and discontinuous aspects in a natural way. 

The particle model for mechanical problems was first used in geomechanics by Cundall [10] in 1979. Later, it was treated in a general and detailed way by Oñate et al. [11].

Currently, most of the literature concerning DEM is performed in geomechanics and rock mechanics. However, the method is rarely used to study additive manufacturing processes, especially powder-based, despite the suitability of DEM for the SLS process.

Figure 2 shows the variables associated with the mechanics of contact between particles in a powder bed. Each particle is described by cartesian components (x,y,z) representing its center of mass, and the vector describing the coordinates of a particle i is denoted by $p_{i}$. The corresponding particle velocity, acceleration, and force vectors are represented by $v_{i}$, $a_{i}$, and $f_{i}$, respectively. The normal and tangential directions are denoted by $\vec{n}$ and $\vec{\tau }$, respectively, and $\delta_{ij}$ represents the overlap between particles i and j.

Unlike other authors in [12,13] who utilize more complex geometries (ellipses and polyhedra), we used spherical particles, as shown in Figure 2. Spherical particles are preferred because of their straightforward geometry, which is identifiable and can be established using just two parameters: the radius and the particle center’s coordinates. 

Additionally, because particle contact detection configurations are straightforward, the approximation is efficient in terms of memory usage and computing code. Because the stiffness of the contact between particles depends on the collision curvature and the particles are vulnerable to rotations that can alter their behavior in the simulation, this sphere/disc geometry is not usually used. Therefore, it is important to apply this approximation carefully.

The main conceptual element of the discrete element method is based on three steps, as shown in Figure 3.

### 2.1. Preliminary Steps

The geometry of the studied medium is represented by hundreds, thousands, and millions of discrete particles filling the modeled space. This geometry forms the simulation domain according to each application or use, such as the representation of bulk materials in tanks, containers, soils, and rocks. 

Once the discrete elements form the geometry of the medium, we need to define all the necessary characteristic parameters that provide information about the geometry and all the material properties.

We also need to select the time step, taking into account several criteria:The precision of the simulations: a small-time step is essential for precise and detailed solutions.The duration of the simulation: if the simulation is too long, a larger time step is required.The influence of the chosen integration scheme: explicit schemes are most commonly used in the literature, and the critical time step limits the selected time step.

### 2.2. Contact Search

The search for contact between particles is the most critical and challenging part of discrete element modeling in terms of the computational effort (60% to 90% of simulation time), especially for non-spherical particles. For this, we need a strategy to manage the set of contacts since it is impossible to perform a collision test for each particle with all other particles in the simulation domain.

In the literature, several techniques exist to spatially partition the domain so that collisions are defined by a neighborhood approach to minimize the number of collision tests. The most common strategies for spatial partitioning include a grid-based method, a tree-based algorithm (K-D-tree, quad-tree, and oct-tree), and a non-binary search method. Recommended references for collision detection methods are [14,15,16,17].

### 2.3. Contact Laws

#### 2.3.1. Mechanical Contact Law

Once the particle collision has been confirmed, the proper contact relationship must be identified in order to record the material’s constitutive response. Figure 2 illustrates how the discrete numerical model calculates the contact force using only the normal and tangential displacement.

Generally speaking, the formulations that describe the typical interactions between the spheres are as follows:

The simplest form of the normal force is for the classical linear spring (Hooke’s law):(1)$$f_{n_{ij}}=-K_{n}.\delta_{ij}.\vec{n},K_{n}=const$$
where $K_{n}$ is the normal stiffness coefficient, $\vec{n}$ is the normal unit vector connecting the centers of the particles. Where $\delta_{ij}$ represents the overlap between the two spheres defined in Figure 2, i.e., the interpenetration distance between particles i and j, calculated using the following formula:(2)$$\delta_{ij}=R_{i}+R_{j}-D_{ij}$$

With $D_{ij}=\left\|p_{i}-p_{j}\right\|$, and $\left\|.\right\|$ representing the Euclidean norm of any vector. In addition, $R_{i}$ et $R_{j}$ are the radii of particles *i* and *j*, and $p_{k}$, k ∈ {*i*, *j*}, is the position vector of the center of mass of the kth particle.

A Hertzian formulation of the normal force between two spheres: (3)$$f_{n_{ij}}=-K_{n}.\delta_{ij}^{\frac{3}{2}}$$
$$with:K_{n}=\frac{4}{3}.\hat{E}.\sqrt{R},\hat{E}=\frac{E}{1-\vartheta }$$
where $K_{n}$ is the normal stiffness coefficient, $\vec{n}$ is the normal unit vector connecting the centers of the particles, and $\delta_{ij}$ represents the overlap between the two spheres defined in Figure 2, i.e., the interpenetration distance between particles i and j, where $\delta_{ij}=R_{i}+R_{j}-D_{ij}$, with $D_{ij}=\left\|p_{i}-p_{j}\right\|$, and $\left\|.\right\|$ represents the Euclidean norm of any vector. In addition, $R_{i}$ et $R_{j}$ are the radii of particles *i* and *j*, and $p_{k}$, k ∈ {*i*, *j*}, is the position vector of the center of mass of the kth particle.

For the shear force, we have the simple formulation used by [7], where the tangential force is a function of the normal force:(4)$$f_{t_{ij}}=-\mu_{c}\left\|f_{n_{ij}}\right\|\frac{\left(v_{i}-v_{j}\right)}{\left\|v_{i}-v_{j}\right\|}$$
with $\mu_{c}$ as a Coulomb friction coefficient, $v_{i}$ and $v_{j}$ are the velocity vectors of particles i and j participating in the collision.

In a powder bed containing multiple particles, each particle may be in contact with numerous particles simultaneously. Therefore, the resulting force on the particle i is written as follows:(5)$$f_{i}=\sum_{j}{(f}_{n_{ij}}+f_{t_{ij}})$$

According to [18], we can simplify the effect of the tangential force in the case where the average diameter of the particle is less than 50 μm and the velocity is low, so the equation of the force applied on the particle *i* becomes:(6)$$f_{i}=\sum_{j}f_{n_{ij}}+g$$
where *g* is the gravitational force.

#### 2.3.2. Thermal Contact Law

In addition to the mechanical generation, the particle system is also engaged in thermal transfer, as shown in Figure 4, which is the exchange of heat between particles.

The heat transfer in the case of particle-particle contact can be considered as pure conduction [7], and neglecting radiation or convection effects, in this case the heat flow from particle j to particle i has the following form:(7)$$q_{ij}=\lambda_{c}\left(T_{j}-T_{i}\right)$$
where $T_{i}$ and $T_{j}$ are the temperatures of particles *i* and *j*, respectively, and $\lambda_{c}$ is a thermal conduction coefficient between particles. This coefficient necessarily depends on the particles’ physical properties, size, and interpenetration distance. 

For particles located on the upper surface of the powder bed, the effects of convection and radiation from the ambient air are added to the total heat flow on these particles. 

### 2.4. Time Step Integration

After determining the thermal and mechanical influences for each particle in the system, it is necessary to propagate these influences in the motion of the particle system and increase the DEM simulation over time.

For discontinuity simulations, explicit time step integration is most frequently utilized in the literature. As demonstrated by [19], who suggested performing a comparison analysis of the most popular explicit time integration schemes in order to provide a clear overview of these schemes and select the best integration scheme for discontinuity modeling.

## 3. Methodology of the Model

Based on the general conceptual elements of DEM from the previous section, a numerical framework based on the discrete element method for powder-based additive manufacturing is developed to model the surface of a polyamide 12 powder layer thermally [8,9]. The details of this framework, the constitutive algorithm, and their implementation are described in the following sections.

### 3.1. Particle System Definition

In this simulation, we consider a DEM particle system of N spherical particles uniformly distributed, as shown in Figure 5, with cartesian spatial coordinates (*x*, *y*) occupying a surface S ∈ $R^{2}$ with bounding planes. A position vector $p_{i}$ represents each particle in the system, which describes the particle’s center of mass. 

There are also other parameters associated with each particle in the domain, $f_{i}$ and $v_{i}$ are, respectively, the corresponding vectors of the force and velocity of particle *i*, with its mass $m_{i}$, radius $r_{i}$, while $T_{i}$ and $q_{i}$ represent the temperature and heat flow. The direction of the parameters during the contact of the particles is well shown in Figure 2.

We simulated the laser projection onto the domain S, defined as a bounded area with 0 ≤ *x* ≤ 3 mm and 0 ≤ *y* ≤ 1 mm, with a uniform average radius over all particles, $r_{i}$ = 0.025 mm, based on the current distribution given by [20] for PA12 powder used in additive manufacturing, as shown in Figure 6, where the average diameter of the particles is 60 μm. We chose a diameter of 50 μm to better distribute the particles in the field and agree with the surface’s dimensions. 

The simulation time variable is denoted τ. At τ= 0, the laser spot is fringing the simulated domain, where the particle system is predefined and its deposition time in S is neglected. The simulation time step is ∆τ, and $n_{\tau }$ is the total number of these time steps in the simulation. For stability, an approximation of the maximum time step given by [7], which requires ∆τ ≪${∆\tau }_{crit}$ with ${∆\tau }_{crit}\sim \sqrt{\frac{m_{min}}{k_{n}}}$. where $m_{min}$ is the minimum particle mass and $k_{n}$ is the normal stiffness used for the calculation of the normal force.

### 3.2. Contact Search

Since the number of particles in the present study is not very high (1200 slices), we saw no need to use any of the previously mentioned algorithms for contact testing. Therefore, contact testing was performed for all particles in the computational domain. To minimize the simulation time and avoid excessive calculations, we added a general condition to stop the contact check when the maximum number of contacts was reached [21,22,23,24,25,26].

The indices of particles in contact with the particle under test are all stored in a separate list labelled $L_{i}$ to calculate the thermal and mechanical contact laws (see Section 3.3).

Two particles are considered to be in contact if the distance between them is smaller than the collision radius. This distance is calculated sequentially according to the following equation:$$\left\|p_{i}-p_{j}\right\|\leq r_{i}+r_{j}$$
where $\left\|p_{i}-p_{j}\right\|$ is the normalized Euclidean distance between the particles in the test, and $r_{i}+r_{j}$ is the sum of the radii of these particles.

### 3.3. Contact Modeling

Once we have detected the collision indices for each particle in the system, we must apply the contact law to the particle system.

#### 3.3.1. Inter-Particle Contact Law

The most straightforward approach described above in Equations (1) and (4) is used to calculate the normal and tangential contact forces. Then, the total force of particle i generated by contact with particle j is calculated in the following form:(8)$$f_{i}=\sum_{j=1}^{n_{c}}\left[f_{n_{ij}}+f_{t_{ij}}\right]$$
when $f_{n_{ij}}$ and $f_{t_{ij}}$ have the form of Equations (1) and (4), and $n_{c}$ is the number of contacts on a particle i.

In addition to mechanical forces, there are also thermal energy transfers. We will model these transfers between particles as pure conduction and neglect the radiative and convective transfers between them. Then, the total heat flow ${(q}_{i})$ on particle i is the sum of all the flows of particles j in contact with i, where it is calculated as follows:(9)$$q_{i}=\sum_{j=1}^{n_{c}}q_{ij}\;with\;q_{ij}=\lambda_{c}\left(T_{j}-T_{i}\right)$$

In this section, we neglect the effect of the boundary planes on the particle system, so we have to add the impact of these planes, which will be discussed in the next section.

#### 3.3.2. Contact Law with the Boundary Planes

Our simulation introduces a rigid stationary plane as a bounding plane. The interaction of the particles with this bounding plane undergoes a normal force given by [7], with a distance condition that must be greater than the particle radius $r_{i}$.
(10)$$f_{n_{i,k}}=k_{b}x_{i}\vec{n_{k}};\;x_{i}=\left\{\begin{array}{c} 0\;if\;\delta_{b}\geq r_{i}\\ r_{i}-\delta_{b}\;if\;\delta_{b}<r_{i} \end{array}\right.$$
where $k_{b}$ is the boundary stiffness, and $\delta_{b}$ is the distance between particle *i* and the boundary plane, $\vec{n_{k}}$ is the normal unit vector to the boundary plane.

These boundary planes also participate in the heat transfer in the powder bed, which generally dissipates the system’s thermal energy to the outside. Then, the heat flow of particle *i* in contact with these planes is regulated by adding a flow by conduction toward these boundary planes, which describes the following formula:(11)$$q_{i}+=\lambda_{cb}\left(T_{b}-T_{i}\right)$$
where $T_{b}$ is the temperature of the bounding plane and $\lambda_{cb}$ is the heat conduction coefficient with the boundary. 

The particles on the upper surface lose heat by convection and radiation to the outside, so the heat flow should be adjusted as follows:(12)$$q_{i}+=h_{c}\left(T_{a}-T_{i}\right)+\zeta .\sigma_{SB}.\left({T_{a}}^{4}-T_{i}^{4}\right)$$
where $T_{a}$ is the ambient temperature, $h_{c}$ is the convective heat transfer coefficient, ζ is the emissivity of the material, and $\sigma_{SB}$ is the Stefan–Boltzmann constant.

### 3.4. Energy Deposition

In our work, we will simulate a stationary projection of the laser onto a bed of PA12 powder with the laser beam ($\beta_{l}$) located at the center of the simulated surface, with coordinates $\beta_{l}=$ (x = 0.0015, y = 0.0005), and a heat input radius $r_{l}$ = 0.0002 m. A view of the simulated granular medium is shown in Figure 7.

The position of the heat source is fixed and known at each time step, so a model describing the flow distribution must be integrated to determine the thermal energy transferred by the laser beam, to each particle, in whole or in part. The area scanned by the laser is often modeled in the literature by a cylindrical heat flow, where the power is constant at all points in the scanned circle. 

In general, the cylindrical heat flow can be described as follows: q0=Pπ*rl, where P is the laser power and $r_{l}$ is the radium of the laser beam. 

The distance between each particle and the laser beam center is calculated at each time step. Depending on the calculated distance, we distinguish between fully or partially scanned particles by the laser (see Figure 8). Additional heat flow is applied to these particles according to:(13)$$q_{i}+=\left\{\begin{array}{c} q_{0}.\frac{S_{i}}{S_{l}},0\leq \left\|p_{i}-\beta_{l}\right\|\leq r_{l}-r_{i}\\ q_{0}.\frac{S_{inter}}{S_{l}},r_{l}-r_{i}\leq \left\|p_{i}-\beta_{l}\right\|\leq r_{l}+r_{i}\\ 0\;\mathrm{otherwise} \end{array}\right.$$
where $\left\|p_{i}-\beta_{l}\right\|$ is the distance between particle i and the laser beam center, $S_{i}$ is the area of particle i, $S_{l}$ Sl is the area sintered by the laser, $S_{inter}$ is the area of intersection between the particle and the heat input area (e.g., the red area in Figure 8), and $q_{0}$ is the total flow delivered by the laser, which is distributed among the particles using the coefficient $\frac{S_{i}}{S_{l}}$ or $\frac{S_{inter}}{S_{l}}$, depending on the degree of exposure of each particle.

### 3.5. Time Step Integration

Based on the integration schemes treated by [19] for the discrete element method, we decided to simulate our problem with the explicit integration scheme of Euler for the evolution in time. We apply the update scheme at each time step after the sequential computation of the forces and the heat flows.

The update scheme for each particle i is expressed as follows:$$v_{i}(\tau +∆\tau )=v_{i}(\tau )+\frac{f_{i}(\tau )}{m_{i}}∆\tau \;p_{i}(\tau +∆\tau )=p_{i}(\tau )+v_{i}\left(\tau \right)∆\tau \;T_{i}(\tau +∆\tau )=T_{i}(\tau )+\frac{q_{i}(\tau )}{m_{i}\times c_{p}}∆\tau$$$c_{p}$ is the specific heat capacity.

### 3.6. Implementation

The flowchart in Figure 9 clearly shows the execution steps followed in the Python implementation for the thermal simulation of the SLS process. The program results are stored for each particle at each time step. The positions of the particles, their radii, and their temperatures are exported and visualized by Python’s visualization library.

## 4. Results and Discussion

We modeled the temperature evolution at the laser beam’s center of PA12 powder, which is typically utilized in the SLS process, using the Python computer language, following the DEM framework for thermal modeling of the SLS procedure (Figure 10).

We compared our findings to an intriguing experimental setup by [8], where the temperature evolution of PA12 particles was examined in a DSC chip with properties similar to the laser used in SLS machines. This comparison served to validate our DEM framework.

Our findings are contrasted with those of [9], who used COMSOL software to predict the PA12 powder bed’s temperature evolution continuously.

All of these assignments were done under the following conditions: The laser power is 1.7 watts, the preheating temperature is 170 degrees Celsius, and all other circumstances are identical to those in Table 1. Due to the minimal difference between the preheating temperature and the sintering temperature, a low-power laser is chosen.

Figure 10 illustrates the temperature increase within the laser spot on the bed surface as a result of our DEM simulation. At the end of the simulation, the temperature is approximately 316 degrees Celsius, which is higher than the liquidus temperature of the PA12 powder, which is 300 degrees Celsius (300 °C). Because of the high input energy (*p* = 1.7 W) and the tiny diameter of the laser beam (200 μm), which led to a high energy concentration on the particles, this simulation generated a significant temperature increase during the duration of the simulation. Because of the continuous loss of thermal energy and the constant laser power throughout the simulations, the figure demonstrates that the progression of the temperature over time is nearly linear. This can be attributed to the fact that the simulations were run with constant laser power.

Our simulation result is compared with the works of [8,9] in Figure 11 to check the agreement of our DEM simulation with these works and evaluate the capabilities of DEM to simulate SLS processes.

Figure 11 demonstrates a strong agreement between our DEM simulation and the experimental result as well as the FEM simulation. The peak temperature attained at the end of the simulations (τ=0.006 s) is very similar in all three simulations, and the temperature evolution is very similar across the three results.

To illustrate the behavior of the PA12 particles in the DEM simulation, Figure 12 shows a view of the temperature evolution for each particle in the region under the laser beam projection for three different time steps. At τ = 0.001 s, the temperature in the powder bed is low, and the distribution is confined to the particles scanned by the laser. 

Due to the low thermal conductivity of the PA12 grains, the medium becomes more homogeneous over time as the temperature in the bed rises. However, this process is gradual because of the heat transmission between the particles and surfaces.

In the second study, we will study the impact of the laser’s power and projection time on the temperature of the domain using the discrete element method. For this, the identical powder (PA12) will be simulated under a static laser source with two different laser powers (0.9 W and 0.6 W), while maintaining the other parameters listed in Table 1.

The results are displayed in Figure 13, which highlights the effect of laser power and scan duration on temperature evolution in the field. Particle temperature increases noticeably, reaching 250 °C at a power of 0.9 W and 200 °C at a power of 0.6 W.

The temperature evolution of the PA12 powder under two different laser powers is compared in Figure 14 with Lanzl’s experimental study [8]. 

The left curve (a) shows the temperature evolution of the PA12 powder from our simulation DEM and the two experimental tests of Lanzl under the same laser power of 0.9 W for different exposure times. 

We can see a good agreement between our simulation and the experimental results since the evolution curve of our DEM simulation always lies between the two tests. 

For a lower laser power, we have the temperature evolution in the right curve (b) with a laser power of 0.6 W. This gives us repeatable results, where the temperature evolution is practically the same for both works with a tiny margin of error.

The exciting comparison in both studies confirms the DEMs ability to model the powder’s thermal behavior under the projection of a laser and demonstrates the accuracy of the discrete element method for modeling the SLS process.

## 5. Conclusions

This study introduces a DEM framework for the thermal simulation of powder-based additive manufacturing processes. A comprehensive framework presents the computations required to characterize the particle interactions in the powder bed. The simulation of a polyamide 12 domain using this framework to predict the evolution of the maximum domain temperature under the static action of a laser point demonstrated its utility for the SLS process.

Using the system depicted in Figure 9, we could monitor the temporal evolution of the PA12 particle temperature. Among the particles of interest were those in the laser range, whose temperature evolution throughout the simulation was reasonably linear. The results were in remarkable agreement with the experimental data collected by Lanzl and the FEM results applied by Yaagoubi on the COMSOL program.

To demonstrate the effects of the laser power and scan length on the temperature evolution of the particle system, a second investigation using this DEM framework was undertaken on the same PA12 particle domain using different laser powers. In addition, we compared these results to Lanzl’s experimental design, which found a comparable level of convergence with a slightly higher margin of error.

These studies demonstrate that our DEM framework can simulate and capture the spatial and temporal thermal distributions of a particle system subjected to the projection of a heat source.

These interesting results will help in the development of a code that incorporates the majority of physical occurrences into the SLS approach, which will be the subject of further research in the near future.

## Figures and Tables

**Figure 1 materials-16-00753-f001:**
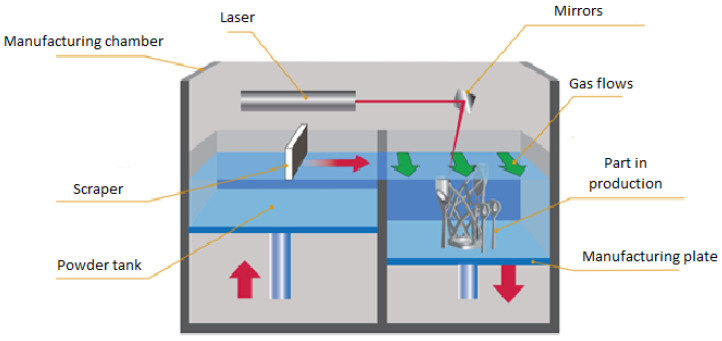
Operating scheme of the SLS process.

**Figure 2 materials-16-00753-f002:**
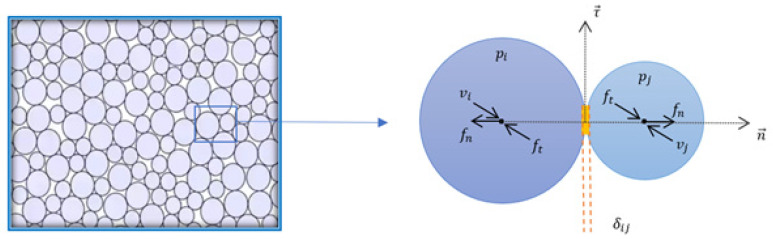
Schematization of the parameters associated with the contact mechanics between the particles.

**Figure 3 materials-16-00753-f003:**
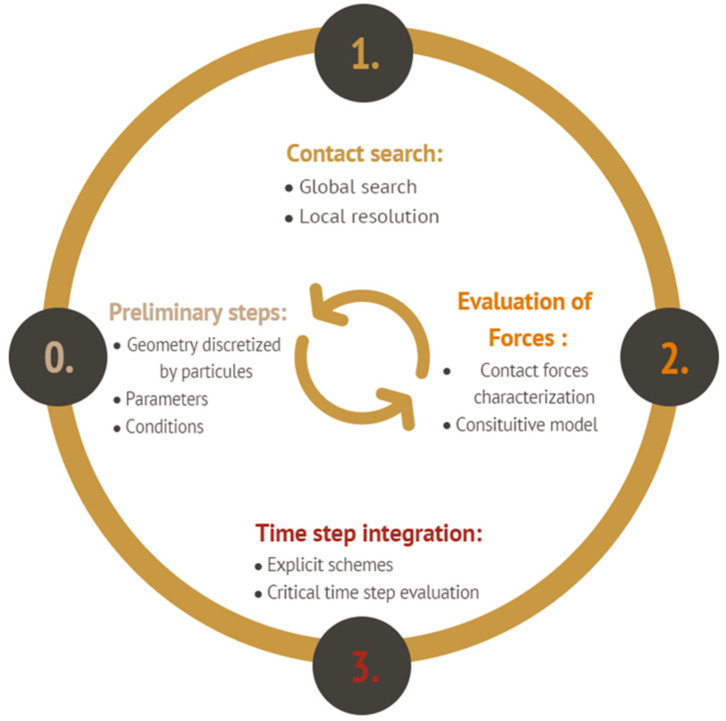
The basic steps for DEM modeling.

**Figure 4 materials-16-00753-f004:**
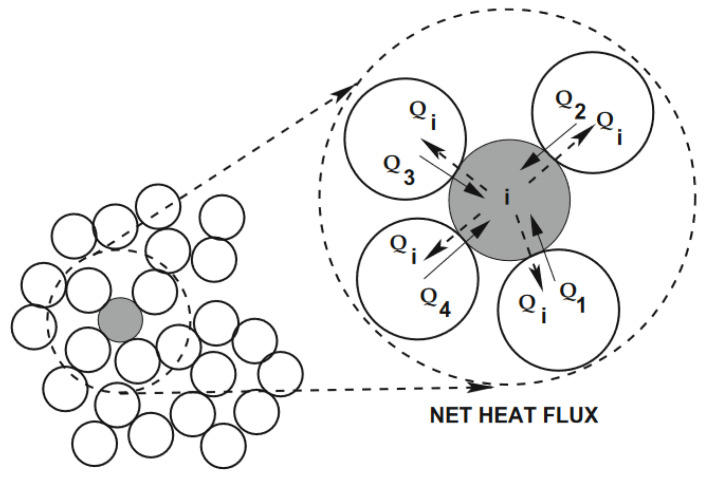
Heat flow exchange between particles.

**Figure 5 materials-16-00753-f005:**
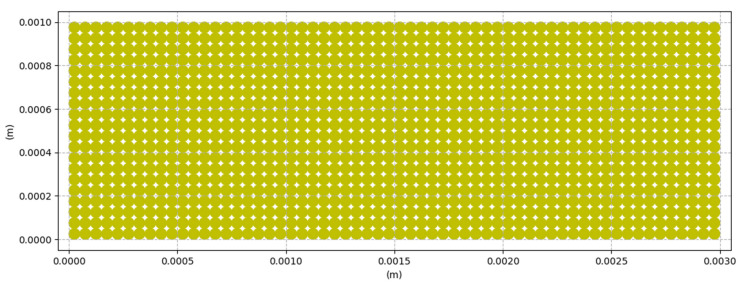
Distribution of DEM particles in the powder bed.

**Figure 6 materials-16-00753-f006:**
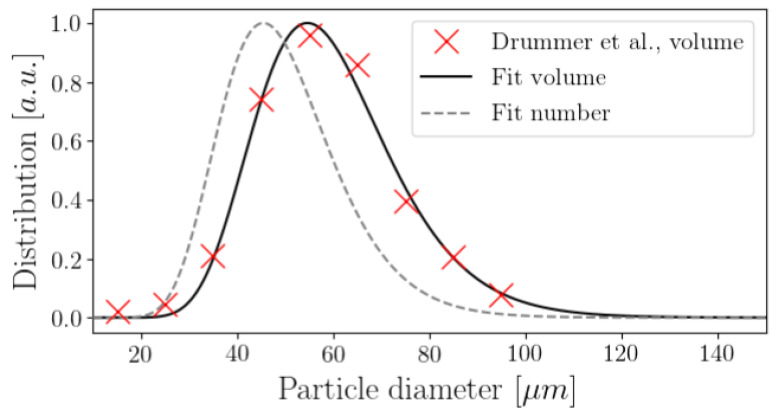
Volume particle diameter distribution density for PA12 powders used in additive manufacturing [20].

**Figure 7 materials-16-00753-f007:**
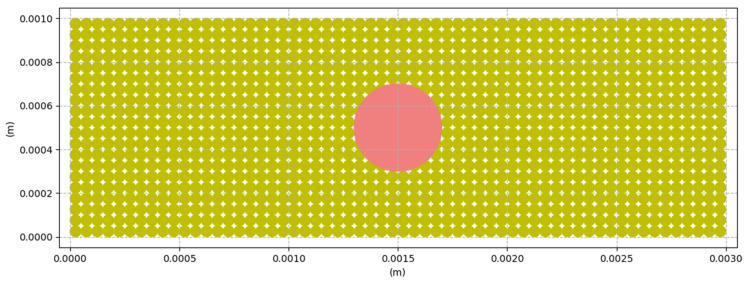
View of the heat input area in our simulation.

**Figure 8 materials-16-00753-f008:**
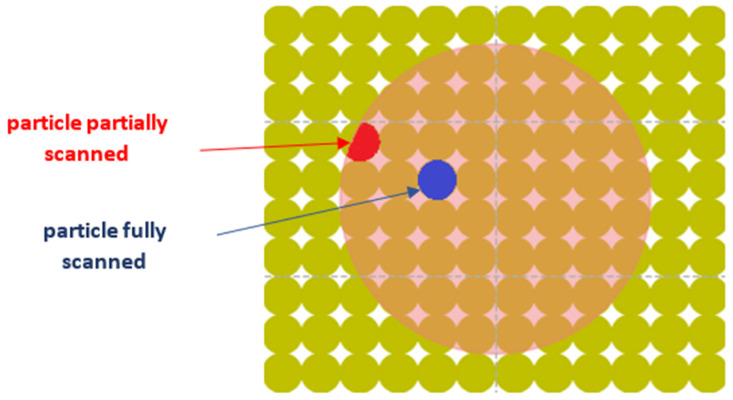
View of the particles completely and partially swept by the laser beam.

**Figure 9 materials-16-00753-f009:**
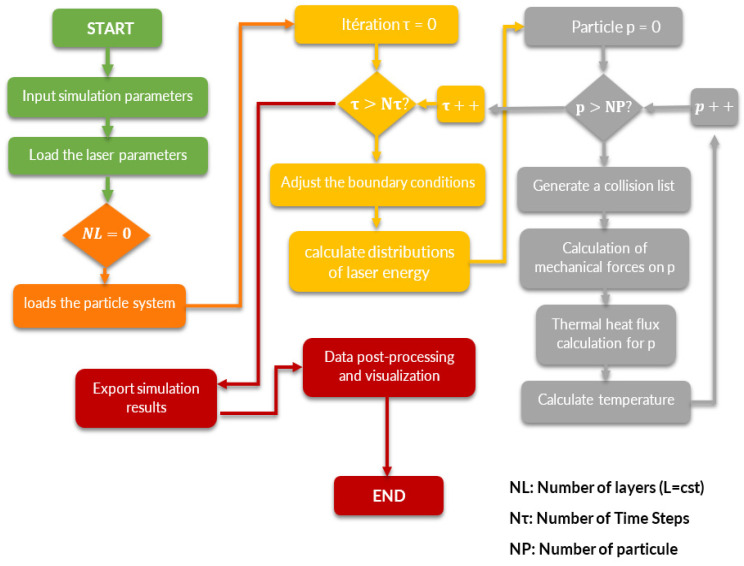
Flowchart in the Python implementation of the DEM framework.

**Figure 10 materials-16-00753-f010:**
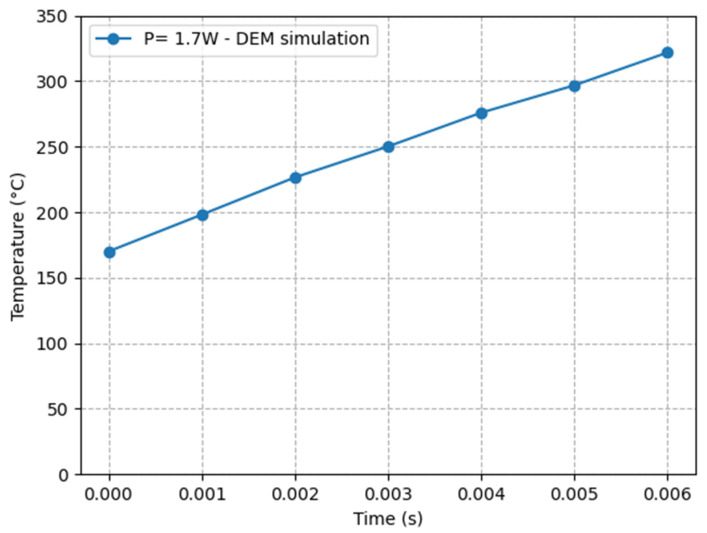
The temperature evolution at the center of the laser beam in our DEM simulations with the parameters from Table 1.

**Figure 11 materials-16-00753-f011:**
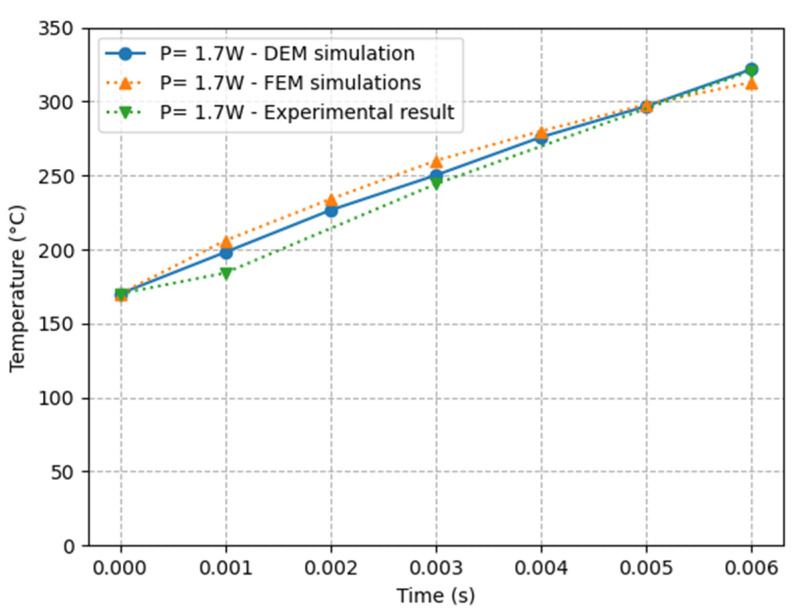
Comparison of the temperature evolution in the center of the laser beam with the works of [8,9] using the same parameters mentioned in Table 1.

**Figure 12 materials-16-00753-f012:**
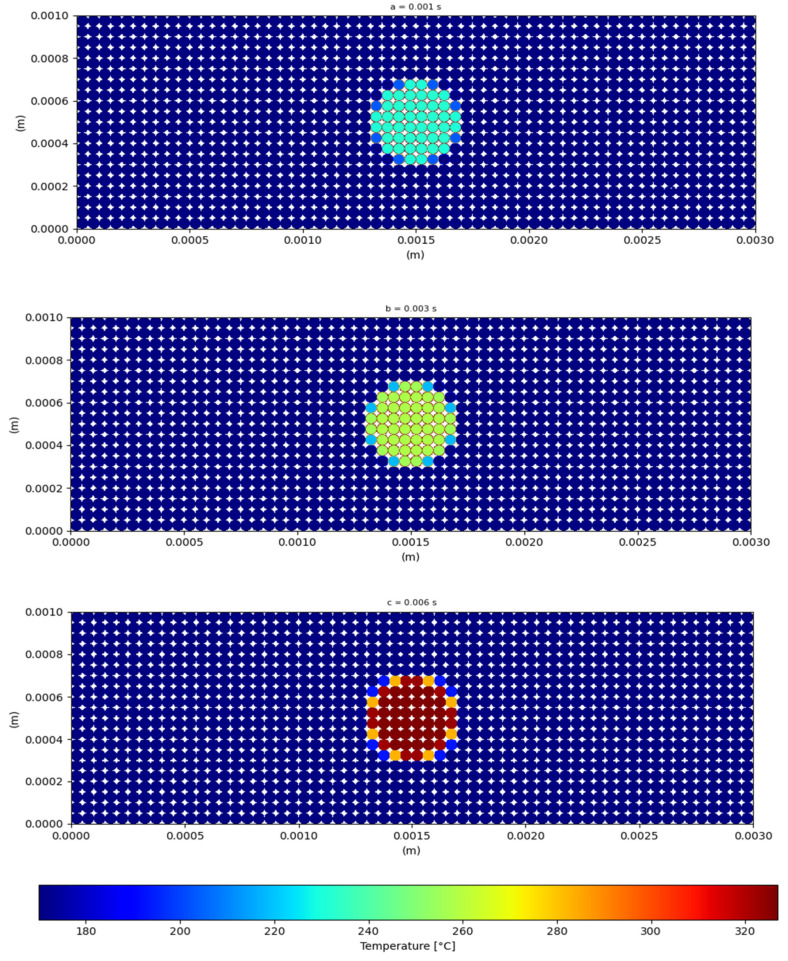
Visualization of the temperature in the powder bed under the effect of a stationary laser at three different times (a = 0.001 s, b = 0.003 s, c = 0.006 s).

**Figure 13 materials-16-00753-f013:**
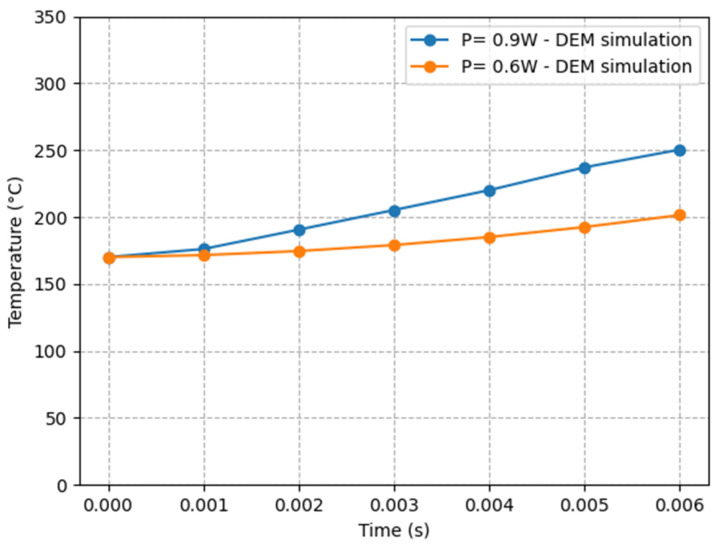
Laser beam center temperature depends on laser scan time and laser power.

**Figure 14 materials-16-00753-f014:**
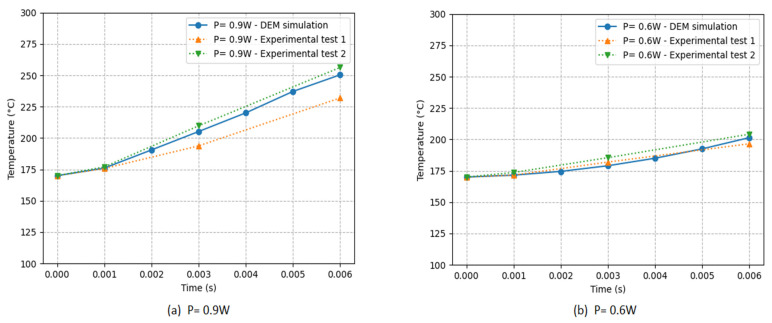
Comparison of the temperature evolution in the center laser beam of our DEM simulation with the [8] results using two different powers (a = 0.9 W, b = 0.6 W) and the data in Table 1.

**Table 1 materials-16-00753-t001:** The values of parameters used in the DEM simulations.

Parameter	Notation	Value
Laser Power	P	1.7 W
Preheating Temperature of the Powder	$T_{i}$	170 °C
Sintering Temperature	$T_{s}$	180 °C
The Temperature of the Chamber	$T_{a}$	170 °C
Laser Beam Radius	$r_{l}$	200 μm
Particle Radius	$r_{i}$	25 μm
Simulation Time	τ	0.006 s
Time Step	∆τ	0.001 s
The Density of The Powder	ρ	1000 $kg/m^{3}$
Thermal Conductivity	λ	0.28 W/m·K
Domain Porosity	ε	0.8 (FEM), 0.78 (DEM)
Stefan–Boltzmann Coefficient	$\sigma_{SB}$	5.67 × $10^{-8}$

## Data Availability

The data that support the findings of this study are available on request.

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
