# Peer review of "Thermal Modeling of Polyamide 12 Powder in the Selective Laser Sintering Process Using the Discrete Element Method"

_materials, 2023, doi:10.3390/ma16020753_

Round 1

Reviewer 1 Report

The present study reports “Thermal Modelling of Polyamide 12 Powder in the Selective Laser Sintering Process using the Discrete Element Method”. To make this paper publishable the authors need to consider following comments.  

-Introduction: It’s a numerical investigation but introduction starts like experimental works with explaining SLS process and not your methods/aims background. In lines 79-82, what is particle physical behavior? And why you think it’s necessary to consider air effect in a vacuum chamber with micron size particles?

-particle radius in Table 1 for nylon PA12 supposed 25 microns; is it uniform or there is a range for simulation?

-Contact search’s sub-steps are missed in figure 3.

This method can be generalized for sintering/melting of metal powders with higher power, higher temperature, and diversity in range of particle sizes?

Generally, paper is written well and everything is addressed. It’s suggested to publish supplementary data with the manuscript to whom want to follow the codes.

Author Response

Dear reviewer, Thanks a lot for your appreciated comment. The suggestion provided by the reviewer is really helpful in improving the overall quality of the manuscript.

Comment 1: Introduction: It’s a numerical investigation but introduction starts like experimental works with explaining SLS process and not your methods/aims background. In lines 79-82, what is particle physical behavior? And why you think it’s necessary to consider air effect in a vacuum chamber with micron size particles?

Authors response: We thank the reviewer for this comment. We have made the necessary changes in the manuscript.

The particle’s physical behavior means that we can use the discrete element method to capture the mechanical and thermal properties of the particles, i.e., the temperature, velocity, and force exerted on each particle in the powder bed.

Considering the vacuum and the effect of the air in the powder bed is intended to make the simulation more realistic since the air between the particles plays an important role, especially in the phase of heat absorption by the powder bed and during diffusion in the medium.

Authors actions: Dear reviewer, we are very grateful for your suggestion. We have updated the introduction section, and described the laser sintering of polyamides with citations from previous works. We have also reduced the part related to the explanation of the SLS process and replaced it with references.

Comment 2:  particle radius in Table 1 for nylon PA12 supposed 25 microns; is it uniform or there is a range for simulation?

Authors response: We thank the reviewer for this comment. A uniform radius of 25 microns was chosen for all particles, corresponding to the volume-particle diameter distribution density in Figure 6.

Comment 3: Contact search’s sub-steps are missed in figure 3.

Authors response:  We very much appreciate the reviewer’s comment. We have made the necessary changes in the manuscript.

Authors actions: Dear reviewer, we are very grateful for your comment. we have added the contact search’s sub-steps in figure 3.

Figure 3: The basic steps for DEM modelling.

Comment 4: This method can be generalized for sintering/melting of metal powders with higher power, higher temperature, and diversity in range of particle sizes?

Authors response: We thank the reviewer for this comment. The method can indeed model sintering and laser melting with different particle sizes. However, our Python code is limited in sintering due to the physical phenomena that are difficult to predict and control in laser melting, such as the occurrence of bubbles, evaporation, decomposition, and change in material properties. This raises difficulties in integrating these phenomena into our code. Nevertheless, the above limitations will be considered in future studies.

Comment 5: Generally, paper is written well and everything is addressed. It’s suggested to publish supplementary data with the manuscript to whom want to follow the codes.

Authors response: We very much appreciate the reviewer’s comment. the article shows the flowchart of the Python implementation with detailed formulations of each step of the DEM framework. We think that we have presented the necessary information to follow the code.

The code will be made available to the researcher once the paper is accepted. We will share the code through GitHub.

Reviewer 2 Report

Dear Authors,

The article is interesting and useful in the development of innovative additive technologies, but it requires a few corrections, which I send attached.

Kind Regards

Reviewer

1.      Line 17: please replace "technique" with "method".

2.      I suggest removing Figure 2 from the Introduction and writing a few sentences about the laser sintering of polyamide powders here and citing several literature references characterizing this technology. The cited reference [1] is good but too general and applies to many other methods as well.

3.      Line 39: The literature please cite.

4.      Line 125: The literature please cite.

5.      Line 128: please explain what the symbols in figure 2 mean, or make this description in the text of the article.

6.      Line 183: I suggest that the explanation of the notations that are given in lines 190-195 be written under equation (1), where dij = Ri + Rj – Dij should be written as a separate equation (2). Whereas equation (2) will then have the number (3).

7.      Line 221: The literature please cite.

8.      Line 223: instead of "we" should be "We".

9.      Line 386 and table 1: please explain in the article why a laser power of 1.7 W was adopted in the simulation since higher power is used in industrial machines, e.g. in the Formiga P100 machine for laser sintering of polyamide powders, the laser power is 30 W.

10.   Line 464: This last conclusion is very optimistic. Perhaps it would be better to write "...most physical events..." instead of "...all physical events...". We don't always know everything.

Author Response

Dear reviewer, Thanks a lot for you encouraging comments. The suggestions provided by the reviewer are really helpful in improving the overall quality of the manuscript.

Comment 1- Line 17: please replace "technique" with "method".

Authors response: We very much appreciate the reviewer’s comment. We have made the necessary changes in the manuscript.

Comment 2: I suggest removing Figure 2 from the Introduction and writing a few sentences about the laser sintering of polyamide powders here and citing several literature references characterizing this technology. The cited reference [1] is good but too general and applies to many other methods as well.

Authors response: We thank the reviewer for this comment. We have made the necessary changes in the manuscript.

Authors actions: Dear reviewer, we are very grateful for your suggestion. We have updated the introduction section, and described the laser sintering of polyamides with citations from previous works. We have also reduced the part related to the explanation of the SLS process and replaced it with references. We have also replaced reference [1] with a reference focused on laser sintering technology.

Polyamide, generally known as nylon, is one of the powders that can be utilized in laser sintering. Polyamide is a durable and flexible material resistant to abrasion and wear. It is utilized in numerous industries, including the automotive, aerospace, and medical sectors. Several academic works aiming to study the SLS process have used PA12 powder as their research subject.

Figure 1 is a simplification of each process involved in the manufacturing of SLS. [1] provides a more in-depth overview of the SLS procedure.

The SLS method results include a variety of defects, such as size faults, layer delamination, porosity, and subpar material qualities. These defects could be visible in the finished products. As a result, in order to achieve the goal of regulating the process-related factors and improving the quality of the final product, it is necessary to have a comprehensive understanding of the behavior of the powder while being projected by the laser source.

Comment 3 : Line 39: The literature please cite.

Authors actions:  Dear reviewer, we are very grateful for your suggestion. We have added the literature citation.

Comment 4: Line 125: The literature please cite.

Authors response: We thank the reviewer for this comment. Line 125 describes what is shown in figure 2, which I created myself.

Comment 5: Line 128: please explain what the symbols in figure 2 mean, or make this description in the text of the article.

Authors actions :  Dear reviewer, we are very grateful for your suggestion. we have added in the text of the article an explanation of all symbols in figure 2.

Figure 2 shows the variables associated with the mechanics of contact between particles in a powder bed. Each particle is described by cartesian components  representing its centre of mass, and the vector describing the coordinates of a particle  is denoted by . The corresponding particle velocity, acceleration, and force vectors are represented by , , and , respectively. The normal and tangential directions are denoted by  and , respectively, and  represents the overlap between particles  and .

Comment 6: Line 183: I suggest that the explanation of the notations that are given in lines 190-195 be written under equation (1), where dij = Ri + Rj – Dij should be written as a separate equation (2). Whereas equation (2) will then have the number (3).

Authors response: We thank the reviewer for this comment. We have made the necessary changes in the manuscript.

Authors actions : Dear reviewer, we are very grateful for your suggestion. We have made the requested change.

The simplest form of the normal force is for the classical linear spring (Hooke's law):

Where  is the normal stiffness coefficient,  is the normal unit vector connecting the centers of the particles. Where  represents the overlap between the two spheres defined in Figure 2, i.e., the interpenetration distance between particles  and , calculated using the following formula:

With , and represents the Euclidean norm of any vector. In addition,  et  are the radii of particles i and j, and ,  ∈ {i, j}, is the position vector of the center of mass of the kth particle.

A Hertzian formulation of the normal force between two spheres: 

Comment 7: Line 221: The literature please cite.

Authors response: We thank the reviewer for this comment. We have made the necessary changes in the manuscript.

Authors actions:  Dear reviewer, we are very grateful for your suggestion. We have added the literature citation.

Comment 8: Line 223: instead of "we" should be "We".

Authors response: We thank the reviewer for this comment. We have made the necessary changes in the manuscript.

Authors actions: Thank you for your comment. We have made the requested change.

Comment 9: Line 386 and table 1: please explain in the article why a laser power of 1.7 W was adopted in the simulation since higher power is used in industrial machines, e.g. in the Formiga P100 machine for laser sintering of polyamide powders, the laser power is 30 W.

Authors actions: Dear reviewer, we are very grateful for your comment. We have made the requested change.

All of these assignments were done under the following conditions: The laser power is 1.7 watts, the preheating temperature is 170 degrees Celsius, and all other circumstances are identical to those in Table 1. Due to the minimal difference between the preheating temperature and the sintering temperature, a low-power laser is chosen.

Comment 10: Line 464: This last conclusion is very optimistic. Perhaps it would be better to write "...most physical events..." instead of "...all physical events...". We don't always know everything.

Authors response: We thank the reviewer for this comment. We have made the necessary changes in the manuscript.

Authors actions: Dear reviewer, we are very grateful for your suggestion. We have made the requested change.

Reviewer 3 Report

A framework of thermal simulation for the selective laser sintering process is proposed based on the discrete element technique. The results showed that the discrete element technique frame could capture the temperature distribution accurately. The reliability of discrete element technique is validated with experiments. The paper is reasonably organized. Nevertheless, the novelty of this study is not described clearly. The editorial support is needed to fix and spell-check the text in general, given that some grammar mistakes and awkward sentences preventing a proper understanding of the content have been found.

Some specific comments relating to parts of the paper are given as follows.

1. It’s better to summarize the limitations of the previous research clearly.

2. Why did you select the discrete element method? What’s the difference between the discrete element method and other method?

3. As shown in Fig. 10, the temperature evolution is approximately kept in Linear. Please interpret this phenomenon.

4. All of the experimental data are from other references. The validation experiments should be provided.

5. For improving the comprehensive frame of the laser processing background, please consider the following citations.

Numerical analysis of the influence of molten pool instability on the weld formation during the high speed fiber laser welding. International Journal of Heat and Mass Transfer, 2020, 160, 120103.

The investigation of molten pool dynamic behaviors during the “∞” shaped oscillating laser welding of aluminum alloy, International Journal of Thermal Sciences, 2022, 173: 107350.

6. The conclusion should emphasize the key points of the current research.

Author Response

Dear reviewer, Thanks a lot for your encouraging comments. The suggestions provided by the reviewer are really helpful in improving the overall quality of the manuscript.

Comment 1- It’s better to summarize the limitations of the previous research clearly.

Authors response: We thank the reviewer for this comment. the limitations of previous studies are all summarized on line (76-83).

However, all cited research works based on continuous methods (FEM, FD, etc.) have limitations when modeling granular media. These models are incapable of capturing the physical behavior of the particles in the powder bed, significantly underestimate the effect of the air between the particles, and ignore the discontinuity of the simulated material.

Moreover, the physical interactions in the powder bed are caused by collisions between the particles. The interactions are highly dependent on the type of bonding between the particles. Due to the SLS process's discrete character and discontinuous behavior, modeling with continuous approaches is incompatible with the procedure.

Comment 2- Why did you select the discrete element method? What’s the difference between the discrete element method and other method?

Authors response: We thank the reviewer for this comment. This is a very important question.

The discrete element method was chosen because the medium modeled in the SLS process is powdery, and DEM is the most suitable method for simulating the mechanical behavior of particle-based systems. The main difference between the discrete element method and continuous methods: In the DEM, the system is represented by a large number of discrete particles, which interact with each other based on their physical properties and the laws of mechanics. This approach is particularly useful for simulating systems with a large number of interacting particles, such as granular materials, powders, and particulate systems. On the other hand, the continuous methods represent the system as a continuous field, discretized into small elements. The system's behavior is then analyzed by solving equations that describe the physical laws governing the field's behavior within each element.

Comment 3: As shown in Fig. 10, the temperature evolution is approximately kept in Linear. Please interpret this phenomenon.

 Authors response: We very much appreciate the reviewer’s comment. We have made the necessary changes in the manuscript.

Authors actions:  Dear reviewer, we are very grateful for your comment. The interpretation of the phenomenon shown in figure 10 was added to the article

Figure 10 illustrates the temperature increase within the laser spot on the bed surface as a result of our DEM simulation. At the end of the simulation, the temperature is approximately 316 degrees Celsius, which is higher than the liquidus temperature of the PA12 powder, which is 300 degrees Celsius (300°C). Because of the high input energy (P = 1.7 W) and the tiny diameter of the laser beam (), which led to a high energy concentration on the particles, this simulation generated a significant temperature increase during the duration of the simulation. Because of the continuous loss of thermal energy and the constant laser power throughout the simulations, the figure demonstrates that the progression of the temperature over time is nearly linear. This can be attributed to the fact that the simulations were run with constant laser power.

Our simulation result is compared with the works of [18] and [19] in Figure 11 to check the agreement of our DEM simulation with these works and evaluate the capabilities of DEM to simulate SLS processes.

Comment 4: All of the experimental data are from other references. The validation experiments should be provided.

Authors response : We very much appreciate the reviewer’s comment. In this phase, we focus on the numerical implementation of the SLS process and on the optimization of our Python code. We intend to implement the experimental part in the future to validate our numerical results.

Comment 5: For improving the comprehensive frame of the laser processing background, please consider the following citations.

Numerical analysis of the influence of molten pool instability on the weld formation during the high speed fiber laser welding. International Journal of Heat and Mass Transfer, 2020, 160, 120103.

The investigation of molten pool dynamic behaviors during the “∞” shaped oscillating laser welding of aluminum alloy, International Journal of Thermal Sciences, 2022, 173: 107350.

Authors response: Dear reviewer, we are very grateful for your suggestion. We have added the necessary reference in the manuscript. Also we added other references.

[22] A. Franco, M. Lanzetta, et L. Romoli, « Experimental analysis of selective laser sintering of polyamide powders: an energy perspective », J. Clean. Prod., vol. 18, no 16, p. 1722‑1730, nov. 2010, doi: 10.1016/j.jclepro.2010.07.018.

[23] T. Stichel et al., « A Round Robin study for Selective Laser Sintering of polyamide 12: Microstructural origin of the mechanical properties », Opt. Laser Technol., vol. 89, p. 31‑40, mars 2017, doi: 10.1016/j.optlastec.2016.09.042.

[24] D. Soldner, P. Steinmann, et J. Mergheim, « Modeling crystallization kinetics for selective laser sintering of polyamide 12 », GAMM-Mitteilungen, vol. 44, no 3, p. e202100011, 2021, doi: 10.1002/gamm.202100011.

[25] B. Yao, Z. Li, et F. Zhu, « Effect of powder recycling on anisotropic tensile properties of selective laser sintered PA2200 polyamide », Eur. Polym. J., vol. 141, p. 110093, déc. 2020, doi: 10.1016/j.eurpolymj.2020.110093.

[26] Ai, Yuewei ; Liu, Xiaoying ; Huang, Yi ; Yu, Long, Numerical analysis of the influence of molten pool instability on the weld formation during the high speed fiber laser welding. International Journal of Heat and Mass Transfer, 2020, 160, 120103.

Comment 6: The conclusion should emphasize the key points of the current research.

 Authors response: We very much appreciate the reviewer’s comment. We have made the necessary changes in the conclusion.

Authors actions:  Dear reviewer, we are very grateful for your comment. We have updated the conclusion.

Conclusion:

This study introduces a DEM framework for the thermal simulation of powder-based additive manufacturing processes. A comprehensive framework presents the computations required to characterize the particle interactions in the powder bed. The simulation of a polyamide 12 domain using this framework to predict the evolution of the maximum domain temperature under the static action of a laser point demonstrated its utility for the SLS process.

Using the system depicted in Figure 9, we could monitor the temporal evolution of PA12 particle temperature. Among the particles of interest were those in the laser range, whose temperature evolution throughout the simulation was reasonably linear. The results were in remarkable agreement with the experimental data collected by Lanzl and the FEM results applied by Yaagoubi on the COMSOL program.

To demonstrate the effects of laser power and scan length on the temperature evolution of the particle system, a second investigation using this DEM framework was undertaken on the same PA12 particle domain with different laser powers. In addition, we compared these results to Lanzl's experimental design, which found a comparable level of convergence with a little higher margin of error.

These studies demonstrate that our DEM framework can simulate and capture the spatial and temporal thermal distributions of a particle system subjected to the projection of a heat source.

These interesting results will help in the development of a code that incorporates the majority of physical occurrences into the SLS approach, which will be the subject of further research in the near future.

Round 2

Reviewer 3 Report

The authors have revised the manuscript according to the comments. This paper can be accepted.